# Multi-criteria optimization for planning volumetric-modulated arc therapy for prostate cancer

Jongmoo Park[1], Jaehyeon Park[2], Sean Oh[2], Ji Woon Yea[2], Jeong Eun Lee[3]* *, Jae Won Park[2]* *

**1** Department of Radiation Oncology, Kyungpook National University Chilgok Hospital, Daegu, Republic of Korea, **2** Department of Radiation Oncology, Yeungnam University College of Medicine, Daegu, Republic of Korea, **3** Department of Radiation Oncology, School of Medicine, Kyungpook National University, Daegu, Republic of Korea

☯ These authors contributed equally to this work.
* jelee@knu.ac.kr (JEL); kapicap@naver.com (JWP)

**Data Availability Statement:** All relevant data are within the manuscript and its Supporting Information files.

**Funding:** This work was partially supported by the 2019 Yeungnam University Research Grant,

## Abstract

We aimed to compare the volumetric-modulated arc therapy (VMAT) plans with or without multi-criteria optimization (MCO) on commercial treatment-planning systems (Eclipse, Varian Medical System, Palo Alto, CA, USA) for patients with prostate cancer. We selected 25 plans of patients with prostate cancer who were previously treated on the basis of a VMAT plan. All plans were imported into the Eclipse Treatment Planning System version 15.6, and re-calculation and re-optimization were performed. The MCO plan was then generated. The dosimetric quality of the plans was evaluated using dosimetric parameters and dose indices that account for target coverage and sparing of the organs at risk (OARs). We defined the rectum, bladder, and bilateral femoral heads. The VMAT-MCO plan offers an improvement of gross treatment volume coverage with increased minimal dose and reduced maximal dose. In the planning treatment volume, the $D_{mean}$ and better gradient, homogeneity, and conformity indexes improved despite the increasing hot and cold spots. When implemented through the MCO plan, a steeper fall off the adjacent OARs in the overlap area was achieved to obtain lower dose parameters. MCO generated better sparing of the rectum and bladder through a tradeoff of the increasing dose to the bilateral femoral heads within the tolerable dose constraints. Compared with re-optimization and re-calculation, respectively, significant dose reductions were observed in the bladder (241 cGy and 254 cGy; $p<0.001$) and rectum (474 cGy and 604 cGy, $p<0.001$) with the MCO. Planning evaluation and dosimetric measurements showed that the VMAT-MCO plan using visualized navigation can provide sparing of OAR doses without compromising the target coverage in the same OAR dose constraints.

## Introduction

Prostate cancer is the commonest malignancy in men worldwide, with an estimated 1,600,000 million incident cases and 366,000 deaths reported in 2015 [1]. For localized prostate cancer, the initial management options include external beam radiation treatment (EBRT),

funded by the the Korean government (MSIT). supported by the the National Research Foundation of Korea [NRF-2019M3E5D1A02068143] The funders had no role in study design, data collection and analysis, decision to publish, or preparation of the manuscript.

**Competing interests:** The authors have declared that no competing interests exist.

brachytherapy, radical surgery, androgen-deprivation therapy, or active surveillance. The treatment modality is selected based on the risk stratification, the patient's preference, resource availability, and clinician judgment of physician [2–4].

In patients with clinically localized prostate cancer, EBRT has equal efficacy as radical prostatectomy [5, 6]. In recent years, compared with the older three-dimensional-conformal radiotherapy (RT) technique, technological advances in RT, such as inverse RT planning and image-guided RT, have facilitated non-compromising dose coverage to the prostate while minimizing radiation to the surrounding normal tissues [7, 8]. This advantage allows higher RT doses to be delivered to the target with simultaneous reduction of toxicity and an improvement in the therapeutic index. Volumetric-modulated arc therapy (VMAT) is one of the methods for inverse RT planning wherein radiation is delivered with gantry rotation and concurrent beam shaping through a continuously moving multi-leaf collimator. Thus, VMAT can provide dosimetric quality that is comparable to that of fixed-beam intensity-modulated radiation therapy (IMRT) and has potential advantages, including a shorter treatment time and a reduction in the monitored units [8–10]. Due to the aforementioned distinction, VMAT is widely used in the clinical setting for prostate cancer. To achieve a VMAT plan with guaranteed advantages, the generation of a high-quality VMAT plan should be guaranteed. The quality of the VMAT plan is determined by how the plan meets the clinical goals when planning RT, which includes: the PTV coverage, PTV dose homogeneity, and normal-organ sparing. These three conflicting parameters should be optimized though tradeoffs according to the competing priorities. The use of this process can translate the clinical intention of the physician into a practical treatment plan. However, this strategy involves an iterative plan optimization, which comprises a trial-and-error process that consumes time and effort between the physicians and planners until an acceptable and optimal plan for clinical delivery can be created. Furthermore, quality of plan is influenced by the planner's skill and experience.

Multi-criteria optimization (MCO), which operates with the Pareto-surface of an optimal plan, is a novel optimization method that was developed and has proven efficiency in terms of the dosimetric quality and planning time [11–13]. MCO can provide real-time dosimetric parameters to the physicians and/or planners via the navigation of the ideal dose-distribution Pareto surface and facilitates the selection of the best plan that fulfills the planned goals of treatment. Thus, MCO permits the avoidance of iterative re-calculation, which is time consuming, and helps physicians and/or planners to create more favorable RT plans. Recently, Eclipse (Varian Medical System, Palo Alto, CA, USA), which enables the development of MCO tools through a visualized tradeoff exploration, was released. However, to the best of our knowledge, there are no reports of a VMAT plan for prostate cancer treatment using MCO in combination with this commercial program. Therefore, this study was conducted with the aim to compare the VMAT plans, with or without MCO, in patients with prostate cancer while using a commercial treatment planning system. For the comparison, we assessed the dosimetric parameters of target volumes and surrounding normal tissues.

## Materials and methods

This study was retrospectively conducted at Yeungnam University Hospital and enrolled a total of 25 patients with low-risk prostate cancer who were previously treated with EBRT using a VMAT technique from November 2015 to March 2019. This study was approved by the Institutional Review Board at Yeungnam University Hospital (YUMC 2021-06-008), and informed consent was waived. All patients underwent computed tomography (CT) simulation in the supine position with the arms folded across the chest and a rectal balloon was inserted. An RT dose of 70 Gy in 28 fractions (once-daily dosing of 2.5 Gy) was delivered to the prostate.

Patients were excluded in cases of high-risk prostate cancer, recurrence, or patients who underwent prostate surgery.

## Target definitions

We delineated the clinical target volume (CTV) of the prostate based on CT images; moreover, in patients who underwent magnetic resonance imaging (MRI), we contoured the target volumes by referring to the MRI images. The CTV included the entire prostate. The planning target volume (PTV) was created by an isotropical expansion of 5 mm of CTV, and this excluded the air region occupied by the rectal balloon. The organs at risk (OARs) to be delineated included the rectum, bladder, and bilateral femoral heads. Furthermore, the same dose constraints for OARs were applied in all the VMAT plans.

## Treatment planning and analysis

The VMAT plan with and without MCO was generated using a commercial treatment-planning system (Eclipse, version 15.6, Varian Medical System, Palo Alto, CA, USA). As the initial VMAT plans used for the treatment were generated using a different version of the Treatment Planning System (Eclipse, version 8.6, Varian Medical System, Palo Alto, CA, USA), the following steps were implemented (Fig 1).

All plans were imported into the Eclipse Treatment Planning System version 15.6, and only a recalculation was performed based on copies of the 8.6 plans. Moreover, as the upgraded version of the algorithm can generate better plans, we performed optimization again using the Photon Optimizer Algorithm (PO) that was available in version 15.6. After the PO process, the MCO plan was generated in the commercial clinical MCO program of Eclipse version 15.6. A radiation oncologist developed and confirmed the MCO plan based on the "tradeoff" with regard to the optimal target volume coverage versus sufficient OARs sparing. All the VMAT plans were calculated by the Anisotropic Analytic Algorithm (AAA dose-calculation algorithm). The treatment plan that normalized at least 90% of the PTV would receive 100% of the prescribed dose. The specific dose–volume constraints of the OARs are shown in Table 1.

To compare the target coverage, the minimum dose ($D_{min}$), mean dose ($D_{mean}$), and maximum dose ($D_{max}$) of the GTV; the $D_{min}$, $D_{mean}$, and $D_{max}$; the gradient index (GI); the conformity index (CI); and the homogeneity index (HI) of the PTV were calculated. The GI is defined as the ratio of the volume receiving 50% (V50) of the prescription isodose around the target to the volume of the prescription isodose (PIV) around the target (Eq 1). The CI is defined as the ratio of the PIV to the volume of the target structure (TV) (Eq 2). The HI is defined as the ratio of difference between the minimum dose that covers 5% and 95% of the PTV (D5 – D95) to the prescription dose (Eq 3).

$$\text{Gradient Index (CI)} = \frac{V50}{PIV} \tag{1}$$

$$\text{Conformal Index (CI)} = \frac{PIV}{TV} \tag{2}$$

$$\text{Homogeneity Index (HI)} = \frac{D95 - D5}{\text{prescribed isodose}} \tag{3}$$

For the OARs, the $D_{max}$ and $D_{mean}$ were compared. The parameters that were obtained included the volume (mL) of the bladder and rectum that received 70 Gy (V70), 60 Gy (V60), 40 Gy (V40), 25 Gy (V25), and 12 Gy (V12) dosing. Statistical analyses were performed by

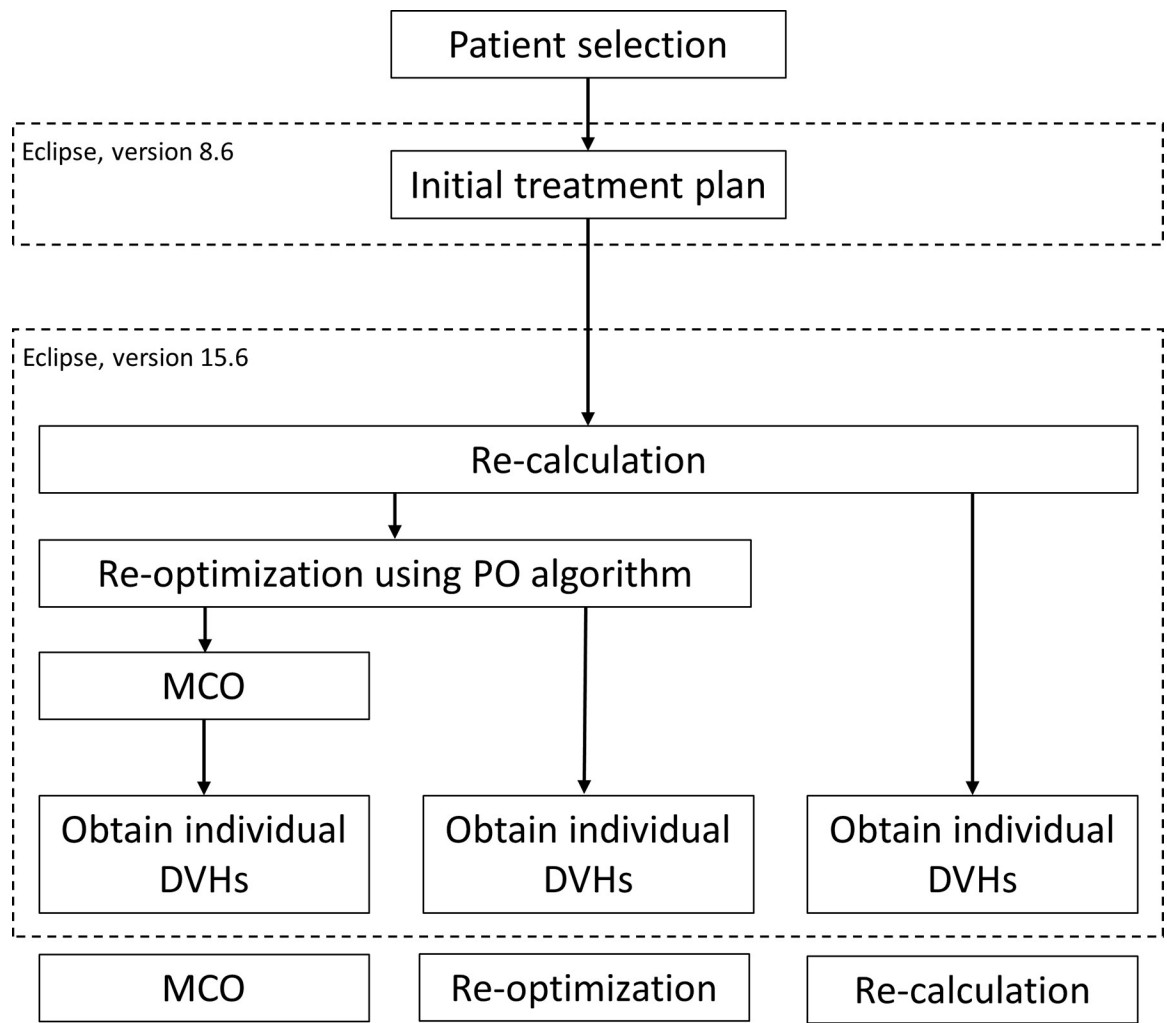

**Fig 1. Dosimteric comparison work flow.** Abbreviations: PO, Photon Optimizer; MCO, multicriteria optimization; DVH, dose–volume histogram.

using the Wilcoxon-signed rank tests to compare the VMAT plans with and without MCO; all statistical analyses were performed using SPSS version 21.0 (SPSS Inc., Chicago, IL, USA).

## Results

A total of 25 RT plans were obtained from patients with prostate cancer who underwent EBRT using the VMAT technique and were enrolled in this study. The median age at which participants in this cohort received an EBRT was 74 years (range 64–83; Table 2). The median volume of the GTV, PTV, rectum, and bladder were 32.50±20.16, 65.90±31.07, 131.90±23.74, and 171.60±149.71 cm$^3$, respectively (Table 2). The average dose–volume histogram (DVH) for all cases are shown in Fig 2, and the dose distribution for a representative case is shown in Fig 3, based on each plan optimization. The DVHs for all individual cases are summarized in S1 Fig.

### Target volume

The dose–volume statistics of the GTV and PTV are summarized in Table 3. In the GTV, the MCO plan led to a small improvement when compared with the re-optimization and re-

**Table 1. Dose-volume constraints for the PTV and OARs.**

|  | Dose volume constraint |
|---|---|
| **PTV** | 90% of PTV with 100% prescribed dose |
| **Rectum** | V70 Gy < 10cc |
|  | V70 Gy < 7% |
|  | V50 Gy < 20% |
|  | V25 Gy < 50% |
|  | V12 Gy < 90% |
| **Bladder** | V40 < 60% |
|  | V60 < 40% |
| **Femur Head** | $D_{max}$ < 50 Gy |
|  | V45 < 10% |
| **Penile bulb** | V40 < 50% |
| **Small bowel** | V35 < 180cc |
|  | V40 < 100cc |
|  | V45 < 60cc |
|  | $D_{max}$ < 50 Gy |

Abbreviations: PTV, planning target volume; OARs, organs at risk; $D_{max}$, maximal dose.

calculation. The MCO significantly reduced the $D_{max}$ from 7363 cGy to 7339 cGy in comparison to the recalculation value. The $D_{max}$ for the re-optimization was reduced from 7363 cGy to 7367 cGy. Further, the MCO increased the $D_{min}$ from 6780 cGy to 6922 cGy in comparison to the recalculation value. On the other hand, the $D_{min}$ for the re-optimization only increased by 4 cGy, in the increment of 6780 cGy to 6784 cGy. These changes improved the $D_{mean}$ for MCO from 7182 cGy to 7122 cGy in the recalculation value, although re-optimization could not improve the $D_{mean}$ (Fig 4).

In the PTV, the MCO plan generated a lower $D_{min}$ from 6481 cGy to 6262 cGy in comparison to the recalculation value. On the other hand, the $D_{min}$ for the re-optimization plan was increased from 6481 cGy to 6471 cGy. With regard to the $D_{max}$, the MCO plan showed more increment than re-optimization in comparison to the recalculation value. The $D_{max}$ for the MCO plan increased from 7394 cGy to 7552 cGy in comparison to the recalculation value; the $D_{max}$ for re-optimization increased from 7394 cGy to 7405 cGy. Nevertheless, MCO improved

**Table 2. Patient characteristics.**

| Variable | Median (±SD) |
|---|---|
| **Age (yr)** |  |
| Median | 74 (±5.44) |
| Range | 64–83 |
| **Target volume** |  |
| GTV | 32.50 (±20.16) |
| PTV | 65.90 (±31.07) |
| **OAR Volume** |  |
| Rectum | 131.90 (±23.74) |
| Bladder | 171.60 (±149.71) |

Abbreviations: SD, Standard deviation; GTV, gross tumor volume; PTV, planning target volume.

**Fig 2. Dose–volume histogram of plans with MCO, with re-optimization and with re-calculation.** MCO, multicriteria-optimization; GTV, gross target volume; PTV, planning target volume; Lt, left; Rt, Right.

the $D_{mean}$. The MCO plan reduced the $D_{mean}$ by 48 cGy, to 7120, compared with 7168 of recalculation. Re-optimization was however increased by 4 cGy, to 7172 cGy. Furthermore, the MCO plans generated better dose gradient, homogeneity, and conformality. However, the GI of the MCO plan did not show a statistically significant difference when compared with the re-optimization plan (1.55 in MCO and 1.67 in re-optimization, $p = 0.65$).

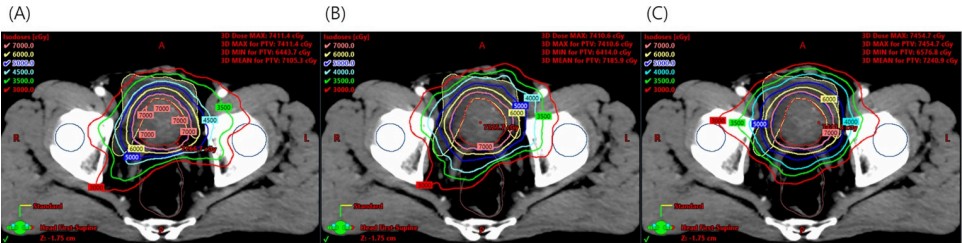

**Fig 3.** The dose distributions for a representative case that were obtained from (A) MCO, (B) re-optimization, and (C) recalculation.

**Table 3. Dose–volume statistics of target coverage with the GTV and PTV.**

| Volume | Dose | MCO | | Reoptimization | | p | Recalcuation | | p |
|---|---|---|---|---|---|---|---|---|---|
| | | Mean | SD | Mean | SD | | Mean | SD | |
| GTV | $D_{mean}$(cGy) | 7122.44 | 51.91 | 7189.44 | 113.34 | <0.001 | 7182.00 | 94.06 | 0.003 |
| | $D_{min}$ (cGy) | 6922.00 | 83.20 | 6784.78 | 80.65 | <0.001 | 6780.11 | 409.95 | 0.001 |
| | $D_{max}$(cGy) | 7339.78 | 111.21 | 7363.11 | 66.38 | 0.020 | 7367.22 | 142.98 | 0.017 |
| PTV | $D_{mean}$(cGy) | 7120.78 | 128.45 | 7172.78 | 67.11 | 0.002 | 7168.44 | 61.20 | 0.002 |
| | $D_{min}$ (cGy) | 6262.56 | 563.29 | 6471.11 | 135.23 | 0.028 | 6481.78 | 102.83 | 0.004 |
| | $D_{max}$(cGy) | 7552.78 | 164.36 | 7405.56 | 87.87 | 0.957 | 7394.44 | 78.22 | 0.864 |
| | V70 (%) | 93.89 | 2.29 | 91.67 | 2.82 | 0.006 | 91.67 | 2.84 | 0.019 |
| | GI | 1.55 | 0.16 | 1.67 | 0.19 | 0.065 | 1.71 | 0.15 | <0.001 |
| | HI | 1.03 | 0.01 | 1.05 | 0.01 | <0.001 | 1.05 | 0.01 | <0.001 |
| | CI | 1.00 | 0.04 | 1.04 | 0.10 | <0.001 | 1.02 | 0.09 | <0.001 |

Abbreviations: MCO, multicriteria optimization; GTV, gross tumor volume; PTV, planning target volume; $D_{min}$, minimal dose; $D_{mean}$, mean dose; $D_{max}$, maximal dose; GI, Gradient Index; HI, Homogeneity index; CI, Conformity index.

## Bladder

In the bladder-dose delivery, MCO achieved significant dose reduction in the $D_{mean}$ by 244 cGy, from 1679 cGy to 1933 cGy, in comparison to the recalculation value. The $D_{mean}$ for the re-optimization plan only decreased by 13 cGy, from 1933 cGy to 1920 cGy. Furthermore, the volumes significantly decreased across all volumes that received 12–70 Gy. However, a dose elevation of the $D_{max}$ of less than 100 cGy was observed in the MCO plan; however, this increase was not significant when compared with the other plans (Fig 5A).

## Rectum

For the rectal doses, the $D_{mean}$ and volume that received 12–60 Gy in the MCO plan showed an improvement over the re-optimization and re-calculation plans (Table 4). A significant reduction of the $D_{mean}$ was achieved in the MCO plan with a decrease of 474 cGy and 604 cGy

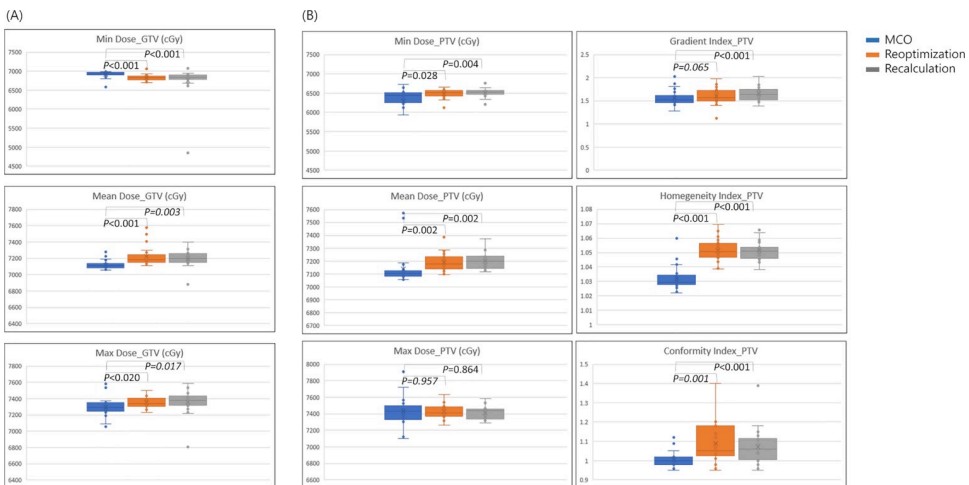

**Fig 4.** Boxplots of (A) $D_{min}$, $D_{mean}$, and $D_{max}$ for GTVs, and (B) $D_{min}$, $D_{mean}$, $D_{max}$, GI, HI, and CI for PTV. Abbreviations: $D_{min}$, Minimum dose; $D_{mean}$, mean dose; $D_{max}$, maximum dose; GI, gradient index; HI, homogeneity index; CI, conformal index.

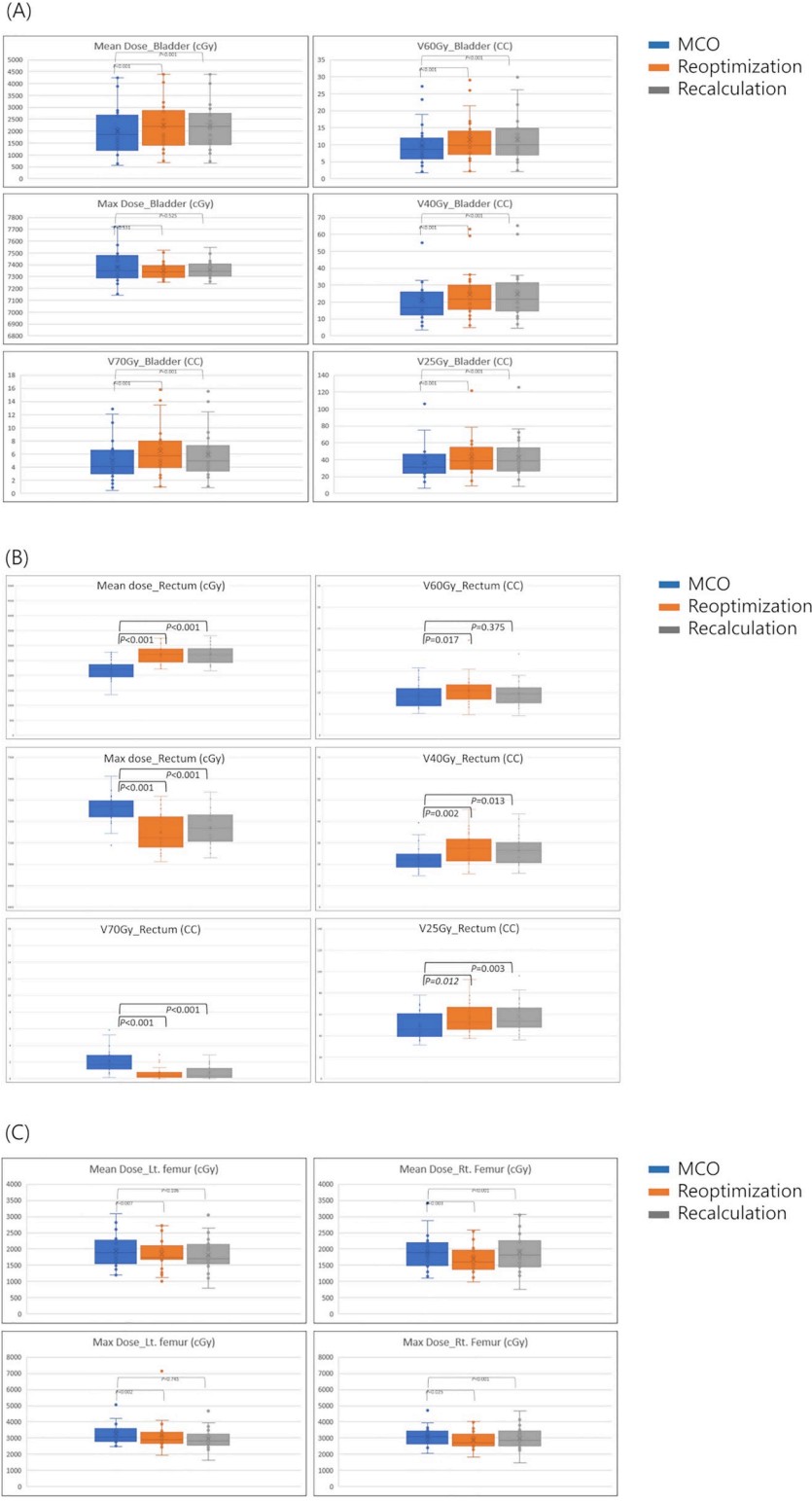

**Fig 5.** Boxplots of $D_{max}$, $D_{mean}$, V70, V60, V40, V25 for (A) bladder, (B) rectum, and (C) $D_{max}$ and $D_{mean}$ for the bilateral femoral heads. Abbreviations: $D_{min}$, Minimum dose; $D_{mean}$, mean dose; $D_{max}$, maximum dose; V(n), volume receiving n Gy.

**Table 4. Dose–volume statistics of OARs with MCO, reoptimization, and recalculation.**

| Volume | Dose | MCO | | Reoptimization | | *p* | Recalcuation | | *p* |
|---|---|---|---|---|---|---|---|---|---|
| | | Mean | SD | Mean | SD | | Mean | SD | |
| **Bladder** | $D_{mean}$(cGy) | 1679.56 | 1028.20 | 1920.44 | 1055.09 | <0.001 | 1933.44 | 1034.67 | <0.001 |
| | $D_{max}$(cGy) | 7420.56 | 140.00 | 7333.11 | 71.93 | 0.331 | 7313.89 | 731.4 | 0.525 |
| | V70 (ml) | 5.20 | 3.28 | 6.55 | 3.81 | <0.001 | 5.88 | 3.74 | <0.001 |
| | V60 (ml) | 10.52 | 6.18 | 12.71 | 6.62 | <0.001 | 12.73 | 6.78 | <0.001 |
| | V40 (ml) | 23.95 | 12.90 | 29.08 | 13.99 | <0.001 | 29.83 | 14.36 | <0.001 |
| | V25 (ml) | 46.36 | 21.13 | 56.49 | 23.35 | <0.001 | 56.44 | 24.83 | <0.001 |
| . | V12 (ml) | 77.05 | 32.80 | 93.46 | 35.71 | <0.001 | 94.64 | 36.53 | <0.001 |
| **Rectum** | $D_{mean}$ (cGy) | 2234.12 | 330.93 | 2708.00 | 299.67 | <0.001 | 2838.11 | 309.06 | <0.001 |
| | $D_{max}$(cGy) | 7265.67 | 66.25 | 7138.00 | 87.13 | <0.001 | 7156.89 | 82.43 | <0.001 |
| | V70 (ml) | 1.71 | 1.40 | 0.47 | 0.80 | <0.001 | 0.59 | 0.81 | <0.001 |
| | V60 (ml) | 9.32 | 2.82 | 12.95 | 3.37 | 0.017 | 9.66 | 2.96 | 0.375 |
| | V40 (ml) | 22.93 | 5.93 | 34.04 | 7.18 | 0.002 | 33.34 | 7.23 | 0.013 |
| | V25 (ml) | 52.21 | 12.70 | 71.38 | 13.84 | 0.012 | 72.60 | 14.62 | 0.003 |
| | V12 (ml) | 84.88 | 15.14 | 91.27 | 15.52 | 0.026 | 91.72 | 15.62 | 0.032 |
| **Femoral head, Rt** | $D_{mean}$(cGy) | 1847.89 | 584.42 | 1643.67 | 425.57 | 0.007 | 1754.67 | 611.27 | 0.106 |
| | $D_{max}$(cGy) | 3049.33 | 548.42 | 2869.89 | 537.88 | 0.002 | 2879.00 | 701.11 | 0.745 |
| **Femoral head, Lt** | $D_{mean}$(cGy) | 1971.11 | 498.66 | 1766.78 | 495.72 | 0.003 | 1833.00 | 513.70 | <0.001 |
| | $D_{max}$(cGy) | 3237.33 | 604.74 | 2976.78 | 967.19 | 0.025 | 2918.22 | 614.37 | <0.001 |

Abbreviations: OAR, organ at risk; MCO, multicriteria optimization; $D_{min}$, minimal dose; $D_{mean}$, mean dose; $D_{max}$, maximal dose; Rt, right; Lt, left.

in the re-optimization and re-calculation plans ($p<0.001$), respectively. However, high-dose regions were compromised. The MCO plan increased the $D_{max}$ by 109 cGy, from 7156 cGy to 7265 cGy, in comparison to the recalculation value. However, the re-optimization plan decreased the $D_{max}$ from 7156 cGy to 7138 cGy. The volume of more than 70 Gy of the MCO plan was 1.71 mL, which was the largest compared with 0.4 mL of re-optimization and 0.59 mL of recalculation ($p<0.001$) (Fig 5B).

### Bilateral femoral heads

In both the femoral heads, a compromising dose distribution was observed. Despite the fact that the MCO plan significantly increased the $D_{mean}$ and $D_{max}$ when compared with the re-optimization and re-calculation plans (Fig 5C), the MCO plan maintained an acceptable increase, which met the dose constraints. The bilateral femoral $D_{mean}$ of the MCO plan increased by more than 200 cGy from the values of the re-optimization plan and by 90 cGy from that of the re-calculation plan to 1847 and 1971 cGy in the right and left femoral heads, respectively. Likewise, both the $D_{max}$ were increased by approximately 300 cGy in the MCO plan compared with those in the re-optimization and re-calculation plans and were measured as 3049 and 3237 cGy in the right and left femoral heads, respectively.

### Discussion

In this study, we used Eclipse 15.6 (Varian Medical System, Palo Alto, CA, USA) to compare the dosimetric parameters and normal tissue sparing in the VMAT planning with and without MCO in the RT plans obtained from the imaging data of 25 patients with prostate cancer. With the same dose constraints, MCO provided better OAR sparing of the rectum and bladder through a tradeoff with the increasing dose to the femoral heads but within an acceptable

deviation while decreasing the minimal point dose and increasing the maximal point dose of the PTV. These sacrifices translated into better dose sparing of the bladder and rectum for the remaining DVHs. Furthermore, this study revealed the results obtained by navigating the competing priorities defined by physicians. With regard to the target dose coverage, the MCO plans offer slightly improved GTV coverage with an increased minimal dose and a reduced maximal dose. In the PTV, an improvement in the $D_{mean}$ as well as better gradient and homogeneity and conformity indices were achieved despite the increasing $D_{max}$ and decreasing $D_{min}$. When implemented through the MCO plan, the plan allowed for a steeper decrease in the adjacent OARs in the overlapping area to obtain lower dose parameters.

In RT for prostate cancer, the surrounding normal tissues, such as the rectum, bladder, and bilateral femoral heads, are considered to be OARs. In particular, the volume of the bladder and rectum frequently overlaps with the PTV depending on the patient's anatomy and PTV expansion that is used [14]. Therefore, RT-related genitourinary and gastrointestinal toxicities are frequently observed, and these can manifest as urinary frequency, dysuria, urgency, rectal pain, stool frequency, or rectal bleeding [15–17]. A recent systemic review and meta-analysis by Carvalho et al. reported that the incidence of acute genitourinary and gastrointestinal toxicity was 31.9% and 21.9% in patients who underwent conventional RT [18]. Moreover, the late genitourinary and gastrointestinal toxicities were 28.0% and 16.2%, respectively. Furthermore, the persistence of radiation-related late toxicities, such as urinary incontinence and rectal discomfort, significantly decreased the quality of life [19]. The risk of toxicity is known to be crucial for the OAR dose–volume parameters [20, 21]. The dose constraints to the OARs remain a major concern, as the related doses can limit the prostate dose levels that are planned. However, higher RT dose levels are associated with improved biochemical tumor control and distant metastasis in patients with localized prostate cancer [22]. Therefore, in practice, reducing the dose that is delivered to the rectum and bladder to the feasibly achievable minimum, given the constraints of the target-dose coverage, is the most important goal of RT planning. The VMAT technique is widely used in prostate cancer and provides a highly conformal target-dose coverage while minimizing the OAR doses [10, 23]. In addition, the optimization of a determinable balance between the target coverage and sparing of OARs is a crucial step for ascertaining the quality of the VMAT plan.

Since Cotrutz et al. initially introduced multi-objective optimization using a tradeoff curve for a case of prostate cancer [24], several studies have shown that the MCO algorithm can be improved. Furthermore, studies have been investigated the effectiveness of MCO for VMAT optimization in prostate cancer [25–27]. Moreover, MCO offers the advantage that the planner and/or physician can select the plan with suitable navigation tradeoffs between conflicting priorities, which represents the most desirable compromise between the OAR DVHs and target-dose coverage. However, the MCO still represents a sophisticated approach for application in clinical practice and requires additional computing resources. Therefore, in the past few years, these limitations have restricted the widespread use of the MCO in clinical practice, and the MCO has only been mainly used in the experimental setting or in institutions where the requisite resource availability.

Recently, the MCO algorithm was implemented in commercial treatment-planning systems, such as RayStation (RaySearch Laboratories AB, Stockholm, Sweden) and Eclipse (Varian Medical System, Palo Alto, CA, USA), and this addressed the difficult issues that are associated with the installation and operation of MCO planning. Several studies have been reported that MCO is a promising and valid optimization technique in RT planning for prostate cancer. McGarry et al. validated MCO in RayStation (v2.4, RaySearch Laboratories AB, Stockholm, Sweden) against a standard step-and-shoot IMRT plan with optimization in Oncentra (v4.1,Nucletron BV, the Netherlands) [28]. The MCO plan showed an equivalent or

better target homogeneity with significant reduction in the rectal dose (30.6±1.4 Gy in MCO and 35.5±4.2 Gy in standard planning; $p$ = 0.047) through a tradeoff of a higher bladder dose within tolerable limits. Additionally, Ghandour et al. reported that, in comparison with the rayArc plan (VMAT), the VMAT-MCO plan generated the same PTV coverage, but with slightly improved OAR sparing on the RayStation [29]. Similar results have been demonstrated using the latest versions of RayStation (version 4.5–6, RaySearch Laboratories AB, Stockholm, Sweden) [30]. Guerrero et al. reported that MCO plans have a slightly better CI and can combine the tumor control probability and normal tissue complication probability models. Furthermore, 15% of patients achieved sparing of the rectal dose. However, another widely used treatment-planning system, the Eclipse, was relatively late to provide an MCO option, and few studies have evaluated the effectiveness of using MCO in Eclipse treatment planning. To the best of our knowledge, this is the first report of an VMAT-MCO plan for prostate cancer that was used in the Eclipse. Our results are consistent with those of previous MCO studies. We demonstrated that the MCO plan, when implemented in Eclipse, can provide OAR sparing, while maintaining the target coverage. Compared with the reports from some previous studies, our results showed greater reductions in the rectal and bladder doses, thereby allowing for higher hot spots, cold spots, and higher femoral head doses than those provided by the standard optimization of treatment-planning systems. This difference may be the result of the fact that MCO more reflects the preference of the physician in the tradeoff between conflicting priorities.

In terms of plan efficiency, to facilitate the use of the MCO, one more process must be added to the workflow of conventional planning, thereby raising concerns about the need for more planning time. However, Gandour et al. reported that the VMAT-MCO plan showed planning efficiency by reducing the planning time whereas proving equal dosimetric quality [29]. Similarly, Müller et al. demonstrated that the physician-derived prostate MCO-IMRT plan achieved better sparing of the rectal and bladder doses and that the MCO work flow did not confer a delay in the planning time constituted by awaiting the physician's approval of the plan [31].

This study has some limitations. First, we analyzed only dosimetric parameters in treatment-planning systems; therefore, it needs to be verified whether a significant OAR dose reduction translates into a toxicity reduction in actual clinical practice. In previous studies, the RT dose was associated with genitourinary and gastrointestinal toxicities [16, 32], and the dosimetric benefits from the MCO plan could contribute to reducing RT-related toxicities. Second, in our study, the evaluation of planning efficiency, such as determination of the planning time and workload, was not performed. Although previous studies reported a maintenance or improvement of the planning time [29, 31], different MCO and treatment-planning system algorithms require validation.

## Conclusions

In conclusion, our results demonstrate that the MCO algorithm in Eclipse 15.6 can improve the OAR doses without compromising the target-dose coverage for VMAT plans in patients with prostate cancer. Further investigations, through a prospective clinical study, are needed to validate whether dose reductions to the bladder and rectum by the MCO planning can result in fewer side effects, with morbidity as an endpoint.

## Supporting information

**S1 Fig. Dose–volume histogram of plans with MCO, with re-optimization and re-calculation for all cases.**
(TIF)

**S1 Data.**
(XLSX)

## Author Contributions

**Conceptualization:** Jeong Eun Lee, Jae Won Park.

**Data curation:** Jongmoo Park, Jaehyeon Park, Sean Oh, Ji Woon Yea, Jae Won Park.

**Formal analysis:** Jongmoo Park, Jaehyeon Park, Jae Won Park.

**Funding acquisition:** Jae Won Park.

**Investigation:** Jaehyeon Park, Sean Oh, Jeong Eun Lee.

**Methodology:** Sean Oh, Ji Woon Yea, Jeong Eun Lee, Jae Won Park.

**Project administration:** Jeong Eun Lee, Jae Won Park.

**Resources:** Ji Woon Yea.

**Software:** Jongmoo Park.

**Supervision:** Jeong Eun Lee, Jae Won Park.

**Validation:** Jongmoo Park, Jeong Eun Lee.

**Visualization:** Jongmoo Park, Ji Woon Yea.

**Writing – original draft:** Jongmoo Park.

**Writing – review & editing:** Jeong Eun Lee, Jae Won Park.

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
