## [Decision Letter · Decision Letter 0]

5 Aug 2021

PONE-D-21-19775

Multi-criteria optimization in volumetric-modulated arc therapy planning for prostate cancer

PLOS ONE

Dear Dr. Park,

Thank you for submitting your manuscript to PLOS ONE. After careful consideration, we feel that it has merit but does not fully meet PLOS ONE’s publication criteria as it currently stands. Therefore, we invite you to submit a revised version of the manuscript that addresses the points raised during the review process.

We look forward to receiving your revised manuscript.

Kind regards,

Henry Woo

Academic Editor

PLOS ONE

“This work was partially supported by the 2019 Yeungnam University Research Grant, funded by the the

Korean government (MSIT). supported by the the National Research Foundation of Korea [NRF2019M3E5D1A02068143]”

 “P.J.W

NRF-2019M3E5D1A02068143

National Research foundation

https://www.nrf.re.kr/index

The funders had no role in study design, data collection and analysis, decision to

publish, or preparation of the manuscript”

6. Please upload a new copy of Figure 3,4 and 5 as the detail is not clear. Please follow the link for more information: " ext-link-type="uri" xlink:type="simple">https://blogs.plos.org/plos/2019/06/looking-good-tips-for-creating-your-plos-figures-graphics/"
https://blogs.plos.org/plos/2019/06/looking-good-tips-for-creating-your-plos-figures-graphics/

7. Please include your tables as part of your main manuscript and remove the individual files. Please note that supplementary tables (should remain/ be uploaded) as separate "supporting information" files.

Reviewers' comments:

Reviewer's Responses to Questions

**Comments to the Author**

1. Is the manuscript technically sound, and do the data support the conclusions?

Reviewer #1: Yes

Reviewer #2: Yes

2. Has the statistical analysis been performed appropriately and rigorously? 

Reviewer #1: Yes

Reviewer #2: I Don't Know

3. Have the authors made all data underlying the findings in their manuscript fully available?

Reviewer #1: Yes

Reviewer #2: Yes

4. Is the manuscript presented in an intelligible fashion and written in standard English?

Reviewer #1: Yes

Reviewer #2: Yes

5. Review Comments to the Author

Reviewer #1: In this manuscript authors compared the VMAT plans, with or without Multi-criteria optimization (MCO), in patients with prostate cancer. For the comparison, authors assessed the dosimetry parameters of target volumes and the surrounding normal tissues. VMAT plan for prostate cancer treatment using MCO is of interest and novel.

Comments to authors:

1. Please explain ORAs in the abstract

2. Authors should perform some histopathology / immunohistochemistry (IHC) to rule out the possibility of surrounding normal tissue toxicities.

Overall the manuscript is well structured. The paper is written very well, and the interpretation of the results is easily understandable. The information provided in this article are of significant interest and novel. I believe that this article is a grave concern to the readership of PLoS ONE after minor revision.

Reviewer #2: My overall impression:

- This is a new study which presumably has not been done before. So this would score good points and would add value to the literature

- The authors need to thoroughly revise the grammar of this manuscript. I would not accept it for publication in the current state

- Need to tidy up the conclusion to link it with the overall aim of the paper which was to compare VMAT plans for prostate treatments etc

- I would accept this for publication pending Major revisions

Specific comments:

Results

Regarding the Results section, they are explaining the values obtained by placing them inside brackets. You don’t do this in a paper.

For example, this is a small section they wrote for the Target volume:

“However, the MCO significantly reduced the Dmax (7399, 7363, and 7367 cGy in MCO, re-optimization, and recalculation, respectively)”

The use of brackets here is confusing to the reader. It cannot make sense which value was obtained with MCO or which value was obtained with re-optimization or which value was obtained with recalculation. They are better off removing the brackets and providing a specific description. Maybe something like this:

However, the MCO significantly reduced the Dmax value from 7339 cGy to 7367 cGy in comparison to the recalculation value. The Dmax for the re-optimization was reduced from 7363 to 7367.

They did the same thing (ie including brackets) for their explanation of the Bladder and Rectum.

They need to write something similar to what they wrote in the Bilateral femoral heads.

Discussion

The discussion is much improved.

Conclusion

I think the conclusion can be improved. This is what I would suggest:

In conclusion, our results demonstrate that the MCO algorithm in Eclipse 15.6 can improve the OAR doses without compromising the target-dose coverage for VMAT plans……etc etc... Further investigation, through a prospective clinical study, is needed to validate that the dose reductions to the bladder and rectum by the MCO planning can result in less side effects with morbidity as an endpoint.

6. PLOS authors have the option to publish the peer review history of their article (what does this mean?). If published, this will include your full peer review and any attached files.

Reviewer #1: **Yes: **ANIS AHMAD

Reviewer #2: **Yes: **Leily Rezaei

---

## [Author Response · Author response to Decision Letter 0]

23 Aug 2021

Dear editor in chief and reviews,

I appreciate your attention and helpful comments which contributed significantly for this

manuscript. I submit the revised manuscript (PONE-D-21-19775) “Multi-criteria optimization for planning volumetric-modulated arc therapy for prostate cancer” to be considered for publication in “Plos One” 

Please find our responses to the reviewer below.

Editor’s Comment

Thank you for your comments. You have made a fair assessment, and we have revised the manuscript to meet the journal’s formatting guidelines..

We attached the minimal data set as an Excel file, as requested.

All relevant data are contained in the manuscript and its “Supporting Information” files.

We have effected the required change, and both titles are now identical..

“This work was partially supported by the 2019 Yeungnam University Research Grant, funded by the the

Korean government (MSIT). supported by the the National Research Foundation of Korea [NRF2019M3E5D1A02068143]”

 “P.J.W

NRF-2019M3E5D1A02068143

National Research foundation

https://www.nrf.re.kr/index

The funders had no role in study design, data collection and analysis, decision to

publish, or preparation of the manuscript”

We have removed all funding-related texts from the manuscript as recommended.

6. Please upload a new copy of Figure 3,4 and 5 as the detail is not clear. Please follow the link for more information: https://blogs.plos.org/plos/2019/06/looking-good-tips-for-creating-your-plos-figures-graphics/" https://blogs.plos.org/plos/2019/06/looking-good-tips-for-creating-your-plos-figures-graphics/

We have uploaded images with higher resolution.

7. Please include your tables as part of your main manuscript and remove the individual files. Please note that supplementary tables (should remain/ be uploaded) as separate "supporting information" files.

We have added tables to the manuscript, and replaced supplementary figures with “supporting information” figures.

Reviewers' comments:

Reviewer #1: In this manuscript authors compared the VMAT plans, with or without Multi-criteria optimization (MCO), in patients with prostate cancer. For the comparison, authors assessed the dosimetry parameters of target volumes and the surrounding normal tissues. VMAT plan for prostate cancer treatment using MCO is of interest and novel.

Comments to authors:

1. Please explain ORAs in the abstract 

Thank you for your recommendation.. We have included a sentence to define OARs, and hope that this sentence provides clarification (line 30). 

The rectum, bladder, and bilateral femora heads were defined as OARs.

2. Authors should perform some histopathology / immunohistochemistry (IHC) to rule out the possibility of surrounding normal tissue toxicities.

We agree that histopathology and IHC of the surrounding normal tissues can indicate the presence or absence of toxicities. However, this study has certain limitations due to its retrospective nature, and patients did not undergo surgery, hence tissues could not have been made available for confirmation.

Overall the manuscript is well structured. The paper is written very well, and the interpretation of the results is easily understandable. The information provided in this article are of significant interest and novel. I believe that this article is a grave concern to the readership of PLoS ONE after minor revision.

Reviewer #2: My overall impression:

- This is a new study which presumably has not been done before. So this would score good points and would add value to the literature

- The authors need to thoroughly revise the grammar of this manuscript. I would not accept it for publication in the current state

- Need to tidy up the conclusion to link it with the overall aim of the paper which was to compare VMAT plans for prostate treatments etc

- I would accept this for publication pending Major revisions

Thank you for providing these insights. We agree with you, and have incorporated all corrections and recommendations made in the whole extent of our study.

Specific comments:

Results

Regarding the Results section, they are explaining the values obtained by placing them inside brackets. You don’t do this in a paper.

For example, this is a small section they wrote for the Target volume:

“However, the MCO significantly reduced the Dmax (7399, 7363, and 7367 cGy in MCO, re-optimization, and recalculation, respectively)”

The use of brackets here is confusing to the reader. It cannot make sense which value was obtained with MCO or which value was obtained with re-optimization or which value was obtained with recalculation. They are better off removing the brackets and providing a specific description. Maybe something like this:

However, the MCO significantly reduced the Dmax value from 7339 cGy to 7367 cGy in comparison to the recalculation value. The Dmax for the re-optimization was reduced from 7363 to 7367.

They did the same thing (ie including brackets) for their explanation of the Bladder and Rectum.

They need to write something similar to what they wrote in the Bilateral femoral heads.

We agree with your assessment. We have rewritten the results to be in line with your recommendations. We hope that the edited section is satisfactory.

Lines 164-178 in the “target volume” section now read,

“The MCO significantly reduced the Dmax from 7363 cGy to 7339 cGy in comparison to the recalculation value. The Dmax for the re-optimization was reduced from 7363 cGy to 7367 cGy. Further, the MCO increased the Dmin from 6780 cGy to 6922 cGy in comparison to the recalculation value. On the other hand, the Dmin for the re-optimization only increased by 4 cGy, in the increment of 6780 cGy to 6784 cGy. These changes improved the Dmean for MCO from 7182 cGy to 7122 cGy in the recalculation value, although re-optimization could not improve the Dmean (Fig 4).

In the PTV, the MCO plan generated a lower Dmin from 6481 cGy to 6262 cGy in comparison to the recalculation value. On the other hand, the Dmin for the re-optimization plan was increased from 6481 cGy to 6471 cGy. With regard to the Dmax, the MCO plan showed more increment than re-optimization in comparison to the recalculation value. The Dmax for the MCO plan increased from 7394 cGy to 7552 cGy in comparison to the recalculation value; the Dmax for re-optimization increased from 7394 cGy to 7405 cGy. Nevertheless, MCO improved the Dmean. The MCO plan reduced the Dmean by 48 cGy, to 7120, compared with 7168 of recalculation. Re-optimization was however increased by 4 cGy, to 7172 cGy.”

Lines 193-195 in the “bladder” section now read,

“In the bladder-dose delivery, MCO achieved significant dose reduction in the Dmean by 244 cGy, from 1679 cGy to 1933 cGy, in comparison to the recalculation value. The Dmean for re-optimization plan only decreased by 13 cGy, from 1933 cGy to 1920 cGy.”

Lines 210-213 in the “rectum” section now read,

“The MCO plan increased the Dmax by 109 cGy, from 7156 cGy to 7265 cGy, in comparison to the recalculation value. However, the re-optimization plan decreased the Dmax from 7156 cGy to 7138 cGy. The volume of more than 70 Gy of the MCO plan was 1.71 mL, which was the largest compared with 0.4 mL of re-optimization and 0.59 mL of recalculation (p0.001) (Fig 5B).”

Discussion

The discussion is much improved.

Conclusion

I think the conclusion can be improved. This is what I would suggest:

In conclusion, our results demonstrate that the MCO algorithm in Eclipse 15.6 can improve the OAR doses without compromising the target-dose coverage for VMAT plans……etc etc... Further investigation, through a prospective clinical study, is needed to validate that the dose reductions to the bladder and rectum by the MCO planning can result in less side effects with morbidity as an endpoint.

Thank you for your suggestion. We have revised the text (lines 314-316) to reflect your recommendations. We think these changes better project the findings of our study.

“In conclusion, our results demonstrate that the MCO algorithm in Eclipse 15.6 can improve the OAR doses without compromising the target-dose coverage for VMAT plans in patients with prostate cancer”.

---

## [Editor Report · Decision Letter 1]

26 Aug 2021

Multi-criteria optimization for planning volumetric-modulated arc therapy for prostate cancer

PONE-D-21-19775R1

Dear Dr. Park,

We’re pleased to inform you that your manuscript has been judged scientifically suitable for publication and will be formally accepted for publication once it meets all outstanding technical requirements.

Kind regards,

Henry Woo

Academic Editor

PLOS ONE

---

## [Editor Report · Acceptance letter]

31 Aug 2021

PONE-D-21-19775R1 

Multi-criteria optimization for planning volumetric-modulated arc therapy for prostate cancer 

Dear Dr. Park:

I'm pleased to inform you that your manuscript has been deemed suitable for publication in PLOS ONE. Congratulations! Your manuscript is now with our production department. 

Kind regards, 

on behalf of

Prof. Henry Woo 

Academic Editor

PLOS ONE